# Why Do They Not Come Home? Three Cases of Fukushima Nuclear Accident Evacuees

**DOI:** 10.3390/ijerph20054027

**Published:** 2023-02-24

**Authors:** Naomi Ito, Nobuaki Moriyama, Ayako Furuyama, Hiroaki Saito, Toyoaki Sawano, Isamu Amir, Mika Sato, Yurie Kobashi, Tianchen Zhao, Chika Yamamoto, Toshiki Abe, Masaharu Tsubokura

**Affiliations:** 1Department of Radiation Health Management, School of Medicine, Fukushima Medical University, 1 Hikariga-oka, Fukushima 960-1295, Japan; 2Department of Public Health, School of Medicine, Fukushima Medical University, 1 Hikariga-oka, Fukushima 960-1295, Japan; 3Health Promotion Center, Fukushima Medical University, 1 Hikariga-oka, Fukushima 960-1295, Japan; 4Department of Internal Medicine, Soma Central Hospital, Okinouchi, Soma, Fukushima 976-0016, Japan; 5Department of Surgery, Jyoban Hospital of Tokiwa Foundation, Fukushima 972-8322, Japan; 6Department of Health Nursing of International Radiation Exposure, School of Medicine, Fukushima Medical University, 1 Hikariga-oka, Fukushima 960-1295, Japan; 7Department of General Internal Medicine, Hirata Central Hospital, Hirata, Ishikawa District, Fukushima 963-8202, Japan

**Keywords:** disaster, returning home, Fukushima nuclear accident, local residents, health issues, aging in place

## Abstract

Many people wish to return to where they used to live after evacuation due to disaster. After the Fukushima nuclear accident in 2011, many residents were forced to evacuate due to concerns about radiation. Subsequently, the evacuation order was lifted, and the government promoted a return policy. However, it has been reported that a considerable number of residents living in evacuation sites or other areas wish to return but are unable to do so. Here, we report three cases of Japanese men and one woman who evacuated after the 2011 nuclear accident in Fukushima. These cases reveal the rapid aging of residents and their health issues. These issues suggest that enhancing medical supply systems and access to medical care can aid in post-disaster reconstruction and residents’ returning.

## 1. Introduction

Living in a familiar place is a universal desire [1,2,3]. This, i.e., the idea of “aging in place”, is also the central concept of life in a local community [4]. Living in a familiar community maintains and improves a person’s independence, dignity, and quality of life [5]. However, appropriate measures are often not adopted when people are affected by disasters and diseases; the individuals affected by such emergencies may be forced to relocate to other places. Being able to respond appropriately to realize “aging in place” in such cases is an important public health issue [6].

The actualization of “aging in place” in the case of a disaster can be challenging for several reasons. Disasters affect the structure and function of families. For example, during long-term evacuation after a disaster, younger generations tend to build new lives in evacuation destinations, while older generations tend to stay on the land where they have lived for many years [7]. The relocation of the younger generation separates a three-generation cohabiting family, leaving only the elderly parents in the village. The residents may experience emotional distress that comes with relocating to a new location [8,9], and there are concerns about the lack of support previously received from family members living together [10]. Additionally, ensuring the quality of life of residents has become a growing national and international concern, because post-disaster communities experience rapid population decline and aging [11]. Countermeasures, such as evacuation and relocation, may be indispensable in the event of disasters. However, a method to realize “aging in place” is still not well established.

The Great East Japan Earthquake occurred on 11 March 2011, when the east area of Japan was hit by a magnitude 9.0 earthquake. A tsunami struck within the next hour, which disabled the power supply of three nuclear reactors (#1, #3, and #4) at the Fukushima Dai-ichi Nuclear Power Plant (FDNPP; operated by the Tokyo Electric Power Company, Incorporated [TEPCO]). With no power supply, the reactors failed to cool down and exploded due to hydrogen generation triggered by the resulting high temperatures. This led to the emission of radioactive nuclides, which were blown in the north-west direction outside the power plants. The Japanese government declared a state of nuclear emergency and ordered the evacuation of residents living within a 30 km radius of the reactors. The residents were immediately evacuated after the evacuation order, which was lifted gradually and partially in the years thereafter. However, to this day, some residents still cannot return to their former residences. Some of them have decided to live in the cities and towns where they evacuated to and settled in after the accident.

Katsurao Village is a mountainous village located 20–30 km away from the FDNPP; its population (1400 in 2011) was evacuated following the announcement of the nuclear emergency. Many residents were evacuated to Miharu Town, Koriyama City, and Tamura City (Figure 1). In the same year, temporary housing was built in Miharu. The populations were scattered, resulting in secondary health impacts [12]. The evacuation order for Katsurao Village was lifted on a large scale in June 2016, and its residents gradually returned home. The current returnee population is 329, which is 25.1% of the village’s registered population (1309 as of 1 December 2022) [13]; the aging rate (65 years and older) of this population is 57%. On the other hand, many residents are still evacuating. Thoughtful measures are needed for the government to implement a return policy in a depopulated area that originally had limited medical resources [14]. Since the lifting of the evacuation order in 2016, the village has adopted various measures to improve the lives of its returnees. However, a certain number of people have decided against returning despite their desire to do so [15,16].

This is a typical report of three individuals who have not returned to their homes in the village ever since the first author of this article has been following up with their health condition after a meeting held for supporting the residents of Katsurao village. While a return policy is being promoted, there are few reports on the actual situation of residents who cannot return. In this respect, this case report can provide information on the current situation and challenges in the reconstruction of life from the perspective of health problems faced by local residents after the nuclear accident.

## 2. Case Presentations

### 2.1. Case #1 Lack of Medical Resources

A man in his 70s whose diabetes worsened during the evacuation period required dialysis treatment three times a week. He used to engage in agriculture, forestry, and livestock farming in Katsurao Village (#1 in Figure 1). After the nuclear accident in 2011, he and his family evacuated to Aizubange Town (#2 in Figure 1) via an evacuation center in Fukushima City (#3 in Figure 1). Subsequently, he moved to the temporary housing built by the village in Miharu Town (#4 in Figure 1). In 2016, he moved into a restoration public housing in Miharu, which was also constructed by the village. In the same year, evacuation orders were lifted on a large scale. However, he did not return to the village despite wanting to do so and decided to continue living in the restored public housing in Miharu. While in Miharu he started hemodialysis, and there were no medical institutions in the vicinity of Katsurao that offered dialysis treatment, so it was easier to go to the hospital in Koriyama from Miharu for dialysis three times a week. He retired his car license for physical reasons. The hospital in Koriyama provides free transportation to Miharu but not to Katsurao, because the distance between the two is too long. Thus, if he had returned to the village, it would have been difficult for him to go to the hospital. It has been more than 10 years since the evacuation and his house in Katsurao was demolished. He told us that he would have liked to return to Katsurao, but he had no choice: it was the best he could do now.

### 2.2. Case #2 Returned Once, but Could Not Stay Long

A man in his 30s returned to Katsurao Village (#1 in Figure 1) once but could not continue living there. He had a physical disability from cerebral palsy and required a wheelchair and assistance with living. After graduating from a support school in Koriyama (#5 in Figure 1), he lived at his home in Katsurao with help from his family. After the nuclear accident, he evacuated to Koriyama and lived in the temporary housing established in Miharu (#4 in Figure 1). During this evacuation outside the village, he began to utilize the daycare services in Koriyama; he commuted to the facility five days a week from Miharu using a day service car. He enjoyed spending every day at the day service. As a result, his quality of life obviously improved. He returned to the village after the evacuation order was lifted in 2016 and used the daycare services in Koriyama while living at home before evacuation. His mother, who was the primary caregiver, died of cancer in 2021. Thereafter, the man did not receive enough care because of the increased care burden on and subsequent fatigue of his father, who was unaccustomed to caregiving. Particularly, driving twice a day to Miharu (a total distance of over 120 km) for a stopover for the day-service commute to Koriyama placed a heavy burden on the father. Moreover, a tumor was found in the father’s brain, and the father required his own medical treatment and living support. The man entered a welfare facility in Koriyama, partly because he wanted to live independently. Currently, his father has recovered from his illness and lives alone in the village, using daycare services three days a week. The man is not permitted to stay overnight at home due to the coronavirus pandemic, but his father drives to meet him once a week or so.

### 2.3. Case #3 Health Problems of a Close Family Member

A woman in her 70s had to stop returning to Katsurao Village (#1 in Figure 1) since her husband entered a facility during the evacuation period. After evacuating to Nihonmatsu City (#6 in Figure 1), she and her husband lived in the temporary housing established in Miharu (#4 in Figure 1). Because her husband’s chronic neurological disease worsened, he entered a nursing facility in Tamura City (#7 in Figure 1). She wanted to visit her husband at the facility occasionally. Therefore, she did not return to her home in Katsurao but built a house in Tamura City, near her husband’s facility, using the reparation money received from the government. Many people had evacuated to Tamura City from Katsurao, and her interactions with them were helpful in her solitary life. Grocery stores and other commercial facilities are within walking distance, meaning shopping is convenient. She stopped returning to the village even after her husband died. She has become accustomed to this life, even though she now has a new house in a new place and lives alone. She is grateful for the occasional visits from old friends. When she must go to the village, she uses the “Otagaisama Taxi” (Otagaisama is Japanese for helping each other) operated by the village, which costs approximately JPY 5000 for a round trip and needs a reservation in advance. She has stated that there is no other alternative to this, even though the fare price is high, because she does not drive. She handed the house in the village to her son.

## 3. Discussion

The aforementioned cases are typical situations that highlight one of the problems in the provision of medical and welfare services in areas affected by the nuclear accident [17]. The impact of a nuclear disaster on community healthcare systems is different from that of other forms of disaster such as earthquake and tsunami. For example, a hospital that residents used to visit was closed due to evacuation orders for several years and not reopened afterwards, which resulted in the loss of accessibility to receive sufficient medical and welfare services continuously after they return to affected areas [18,19]. Hence, it is very difficult for returning residents who need to keep using services such as medical and nursing care to settle down in the affected areas [20]. The evacuation of the, mostly, younger generation to escape radiation resulted in a more aged population in the affected areas, changing the social structure and weakening the community as a whole. There are reports of people in the area skipping regular hospital visits and health checkups due to lack of notification from family members, resulting in the delayed detection of cancer [21].

It became clear that there was a lack of health support that was necessary for people to return and rebuild their lives under the government’s return policy after the nuclear accident [22]. However, it is not realistic to increase medical institutions and welfare services in areas with declining populations. Fortunately, since Katsurao is a small village, the support service there allows for a face-to-face relationship with the residents and stakeholders; this is a positive aspect of this village. It is even more important to build a comprehensive community care system that seamlessly provides medical, long-term, preventive, and livelihood support services after the residents return to the village. The most feasible and practical solution would be to improve the functions of the pre-existing healthcare facilities and create a system of cooperation between the social welfare council and the village office. In fact, in Katsurao Village (where the demand for nursing care has surged since 2011), nursing care measures are needed urgently [23,24], and development focusing on nursing care prevention is essential [25,26].

A return intention survey revealed that 27.7% of Katsurao residents decided not to return to the village [27]. For the residents who did not return, problems with access to medical and welfare services were raised in addition to health problems for themselves and their families [28]. Medical institutions and welfare services are located far from the village despite the rapidly aging village population and the consequent increase in the number of residents who need outpatient treatment. During the prolonged evacuation period, many have already utilized health-related services at their evacuation destinations. The lengthy evacuation period and TEPCO’s compensation for securing housing [29] may have encouraged evacuees to settle in their evacuation destinations. Regular outpatient treatment at a hospital far from the village, with scarce public transportation facilities, increases the burden on patients and their families. While the children and grandchildren’s generation establish their lives connected to schools and jobs at the evacuation site, many elderly people return to their homes in the village alone. In such cases, they cannot expect the informal support from family members living with them that they had previously obtained. Some neighboring residents may help each other and drive to a hospital in Koriyama by car. However, in villages where only the elderly return, continuing such mutual aid is difficult. It is significant to improve the means of public and private transportation.

These areas facing depopulation after the nuclear accident are experiencing some of the current social issues of Japan in advance, such as the problem of isolation, the rapidly declining birthrate and aging population. After five years since the nuclear accident, although the government has adopted a return policy, returning does not smoothly progress and residents face various difficulties when they return. Local municipalities are trying to help and support the residents who are still living outside of the affected area, as well as those who have already returned. However, they cannot promote what actions to perform and provide enough support for them.

Our findings could indicate the possible health issues for residents in areas that need to evacuate or return in future disasters. Through their health challenges, they may provide suggestions to local government officials on the need to improve medical and welfare systems and transportation systems.

## 4. Conclusions

Japan is the first country to adopt a policy of return after the evacuation order along with the nuclear accident. Since the effects of several years of evacuation, although the subsequent lifting of evacuation order was announced in 2016, approximately 70% of the local residents have not returned yet. It is quite possible that it will continue as a small municipality of approximately 300 residents. The same applies to the nearby formerly evacuated areas. By looking at these real cases of residents who wanted to but could not return, we found that the accessibility to sufficient medical and welfare services had worsened for those who will need them if they return.

Health-related support is one of the important social resources for the implementation of the return policy, i.e., for residents to be able to return to their original address of choice at any time and rebuild their lives. The needs of the residents must be identified and prioritized for maintenance while also considering the feasibility.

## Figures and Tables

**Figure 1 ijerph-20-04027-f001:**
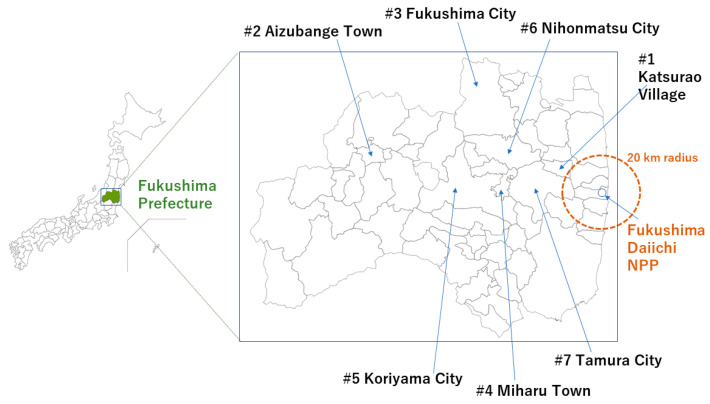
Location of municipalities during the evacuation process from Katsurao Village. The geographical relationships between the Fukushima Daiichi Nuclear Power Plant and Tamura City, Miharu Town, Koriyama City, Nihonmatsu City, Fukushima City, and Aizubange Town are shown.

## Data Availability

Not applicable.

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
