# Peer review of "Why Do They Not Come Home? Three Cases of Fukushima Nuclear Accident Evacuees"

_ijerph, 2023, doi:10.3390/ijerph20054027_

Round 1

Reviewer 1 Report

Thank you for the opportunity to review this case study exploring decision making and experiences of return home among evacuees following the Fukushima nuclear accident. The study is interesting, concise and informative. The manuscript is well written, and highlights important healthcare needs for an ageing population in Japan. The interpretation of the findings is appropriate and sufficiently nuanced. Please see my comments below regarding suggestions for revision.

1.   Introduction: Please define ‘aging in place’ to clarify for readers not familiar with the literature.

2.       Please provide further detail on the meaning and implications of an ‘aging rate of 57%’.

3.       Methods: Please provide detail on how the case studies were chosen. It is important to know how the researchers recruited the three people featured, and the inclusion/exclusion criteria applied for the study?

4.       Methods: How were the case studies conducted?

5.       It is not clear whether ethics approvals were sought and obtained (although I see that written informed consent was obtained).

6.       Discussion: It may be worthwhile exploring the literature on factors associated with decision making regarding return home to rural and isolated areas

7.       There is no mention of radiation concerns in the manuscript. It may be that this wasn’t a concern cited in the three case studies, but it should be addressed given the nature of the disaster.

Minor comments:

1.       Unclear who ‘they’ refers to (line 38)

2.       Case #2 – at what point did the man return to Katsurao Village (first sentence)?

3.       Case #3 – what sort of facility did the husband enter during the evacuation?

Reviewer 2 Report

Dear authors,

Thank you for reporting the information regarding evacuees.

below are my comments:

The information provided on ageing in place and the need to find the reasons for not returning to place were well established. However, I think that the authors should add more information to help guide the readers on why these three cases were chosen to be reported. The authors did use the word typical cases, however, I would like to see more references to the ideas. Moreover, the case report should be either a new or uncommon phenomenon more than the most common or typical ones.

Regarding the case presentation, would it be possible to add more objective information to each case's detail? The information reported looked really subjective and specific which may not reflect the overall lack of health supports issues that the authors trying to make both in the introduction and the discussion. For instance, 'his quality of life improve' (Line 98), The father's burden (Line 101-103), Her husband's chronic neurological disease worsened, and he entered a facility in Tamura City (Line 115), etc.

The discussion seemed to be independent of the case presentations, even though, all points raised in this part were important points to make. Please make the discussion more related to the case presentations.

Best regards,

Reviewer

Reviewer 3 Report

This manuscript, entitled “Why Do They Not Come Home? Three Cases of Fukushima Nuclear Accident Evacuees”, is submitted as a Case Report and assesses the intention of returning to their hometown after nuclear disaster evacuation. I applaud the authors’ efforts to bring this topic up and raises unique characteristics of nuclear disasters compared with “conventional” natural disasters (e.g., earthquakes, tsunamis). Having said that, I have several major/minor comments that might be helpful in improving the quality of this paper.

Major comments

1.    I am unsure if this report can be categorized as “Case reports”. The three cases are not about the individuals’ medication conditions but about decisions on whether to go back to one’s hometown after the nuclear disaster.

2.    This manuscript is too difficult to read without prior knowledge of the Fukushima nuclear disaster. Although I understand that Introduction should be succinct, please give enough background information to the readers who do not necessarily have knowledge of this unique disaster.

3.    The authors neither explain nor discuss the uniqueness of evacuation features between nuclear and other types of disasters. Particularly, these features may include mass evacuation from the affected region in the immediate aftermath, heightened radiation dose, and persisting radiation-related restrictions/concerns.

4.    In a similar fashion, the authors should give a guidance about the impact of the Fukushima nuclear disaster to the regional healthcare systems. How is this trend different from other regions affected by the earthquake and tsunamis but not by nuclear accident?

5.    The authors use the expression “rapid aging of residents”, but this expression can be misleading because, of course, each resident does not age rapidly. Please use other expression that accurately describes the aging trend.

6.    The authors’ discussion points are unclear. What did the authors learn from the three cases and available literature, and how can these lessons be applied to future nuclear disaster preparation? Also, please refrain from suggesting a narrow, specific topic (i.e., discussing village clinic’s function).

Minor comments

1.    Page 2, line 62: Please give a definition for an “aging rate”.

2.    Page 2, line 98: What is “day service”?

3.    Page 3, Figure 1: Please define an acronym “NPP”.

4.    Page 4, line 154, “27.7% of residents”: Please clarify “residents”.

5.    References are not properly formatted.
#1: Author names, journal name
#12, 13: Same reference?
#27: access year
#29: access year

Round 2

Reviewer 2 Report

Dear authors,

Thank you for revising the manuscript.

Reviewer 3 Report

I thank the authors for taking comments from other reviewers and myself into considerations and revising the manuscript accordingly. I have to confess, however, that I have follow-up comments for Comments #2, #4, and #6.

---

2.       This manuscript is too difficult to read without prior knowledge of the Fukushima nuclear disaster. Although I understand that Introduction should be succinct, please give enough background information to the readers who do not necessarily have knowledge of this unique disaster.

    Thank you for informing us. The third paragraph of the Introduction mentions the Fukushima nuclear accident and the location, evacuation status, and population of Katsurao Village. We appreciate your understanding.

The authors are not responding to my query. For example, the third paragraph is so diifficult to read because a reader with no background knowledge has to deal with unknown terms (Katsurao village, Fukushima Daiichi NPP, and various city names) and later find out that this paper is about the Fukushima nuclear disaster. Imagine you are explaining this situation to a person who has no knowledge about nuclear disaster and reorganize this section so that (s)he can gradually hone in to the topic of interest. Do not assume (s)he knows that Fukushima Daiichi NPP is the source of the disaster and that TEPCO operates it. Write about the Fukushima nuclear disaster first, and then gradually add details. If other parts of the paper has this trend, please consider revising them as well.

---

4.       In a similar fashion, the authors should give a guidance about the impact of the Fukushima nuclear disaster to the regional healthcare systems. How is this trend different from other regions affected by the earthquake and tsunamis but not by nuclear accident?

   Thank you for sharing your important perspectives...despite the fact that it has been 6 years since the evacuation order was lifted in 2016, the population has not fully returned to the area. We have also mentioned the issue of medical and welfare supply in the evacuation zone as a factor (first paragraph of the discussion) and the housing guarantees by TEPCO to the evacuated residents (third paragraph of the discussion). These phenomena are very different from those in areas affected by earthquakes and tsunamis alone. We would appreciate your understanding.

Thank you for your comments. In the current form, however, the readers without background knowledge may not understand that it is indeed different. Is it possible to write in the text that the effect to regional healthcare systems from a nuclear disaster is different than that of other forms of disaster?

---

6.    The authors’ discussion points are unclear. What did the authors learn from the three cases and available literature, and how can these lessons be applied to future nuclear disaster preparation? Also, please refrain from suggesting a narrow, specific topic (i.e., discussing village clinic’s function).

    Thank you for sharing your important points. We believe that through the three case studies, we have gained insight into the need for improving the health and welfare system and enhancing access to health and welfare services in areas that will require evacuation and return after future disasters. I have only conveyed what I have learned from walking the frontlines and interacting with professionals in the field about what is needed in the areas after the nuclear accident. Although you may say that improving the village clinic is too specific and narrow discussion, we believe that the medical circumstances improvement is definitely crucial for residents in such a situation Katsurao village is currently facing, which the first author actually found out through her activities.

I understand that this paper’s findings stem from your team’s medical support to the affected region. Still, I find it crucial for the readers to learn lessons from your valuable experiences, because if they ever experience nuclear disasters, they are likely to face similar challenges. Could you at least rephrase and generalize your statements? Rather than emphasizing an importance of improving the functions of the “Katsurao village clinic”, you can mention it as “preexsiting healthcare facilities”. Likely, rather than saying that “village taxi services” and “budgets to improve the existing route bus” are important, how about "improving means of public and private transportation” are important?